# Operative Digital Enhancement of Macular Pigment during Macular Surgery

**DOI:** 10.3390/jcm12062300

**Published:** 2023-03-16

**Authors:** Otman Sandali, Rachid Tahiri Joutei Hassani, Ashraf Armia Balamoun, Alan Franklin, Ahmed B. Sallam, Vincent Borderie

**Affiliations:** 1Centre Hospitalier National d’Ophtalmologie des XV-XX 28, Rue de Charenton, 75012 Paris, France; 2Service de Chirurgie Ambulatoire, Hôpital Guillaume-de-Varye, 18230 Bourges, France; 3Service de Chirurgie Ambulatoire, Centre Hospitalier de Granville, 50400 Granville, France; 4Watany Eye Hospital, Cairo 11775, Egypt; 5Watany Research and Development Centre, Cairo 11775, Egypt; 6Ashraf Armia Eye Clinic, Giza 12655, Egypt; 7Diagnostic and Medical Clinic, 1720 SpringHill Ave Suite 300, Mobile, AL 36604, USA; 8Department of Ophthalmology, Harvey and Bernice Jones Eye Institute, University of Arkansas for Medical Sciences, 4301 W Markham St, Little Rock, AR 72205, USA

**Keywords:** digital visualization, 3D surgery, macular surgery, macular pigment, color filters, epiretinal tissue, contrast, peeling, enhancement

## Abstract

Purpose: To describe the feasibility of intraoperative digital visualization and its contribution to the enhancement of macular pigmentation visualization in a prospective series of macular surgery interventions. Materials and Methods: A prospective, single-center, single-surgeon study was performed on a series of 21 consecutive cases of vitrectomy for various types of macular surgery using a 3D visualization system. Two optimized filters were applied to enhance the visualization of the macular pigment (MP). For filter 1, cyan, yellow, and magenta color saturations were increased. Filter 2 differed from filter 1 only in having a lower level of magenta saturation for the green-magenta color channel. Results: Optimized digital filters enhanced the visualization of the MP and the pigmented epiretinal tissue associated with the lamellar and macular holes. In vitreomacular traction surgery, the filters facilitated the assessment of MP integrity at the end of surgery. Filter 1 enhanced MP visualization most strongly, with the MP appearing green and slightly fluorescent. Filter 2 enhanced MP visualization less effectively but gave a clearer image of the retinal surface, facilitating safe macular peeling. Conclusion: Optimized digital filters could be used to enhance MP and pigmented epiretinal tissue visualization during macular surgery. These filters open new horizons for future research and should be evaluated in larger series and correlated with intraoperative OCT.

## 1. Introduction

High concentrations of macular pigment (MP) are present in the retina. MP consists of xanthophyll carotenoids, which are particularly abundant in Henle’s fiber layer of the central macula and in the inner and outer plexiform layers outside the fovea [1,2].

Light passes through the MP to reach the photoreceptor outer cell segments and retinal pigment epithelium. MP acts as a filter, protecting the macula from blue light, and as a resident antioxidant, decreasing the damage caused by oxidative stress [3]. The normal distribution of MP and several abnormalities of this distribution in macular diseases have been described [4,5].

Aspects of MP disruption have been described in surgical retinal diseases, mostly in vitreomacular traction and epiretinal proliferation associated with lamellar or macular hole diseases. Indeed, Obana et al. recently described the appearance of MP in eyes with different stages of macular holes [6]. They observed a lack of MP within the hole in an area corresponding to an outer plexiform layer defect. Therefore, MP analysis may be of interest in vitreomacular traction surgery for the detection of iatrogenic macular holes.

MP has been reported to be present in epiretinal proliferation associated with lamellar or macular holes [7]. Gentle handling, with preservation of the epiretinal proliferative tissue, is recommended to ensure a successful surgical outcome [8,9]. However, in surgical practice with standard visualization, it may be difficult to obtain a clear view of the MP, the yellowish color of which is close to the orange color of the retina, resulting in low contrast between them.

Digital technology for three-dimensional (3D) visualization has recently been introduced in ocular surgery [10,11,12,13]. With this technology, digital filters can be applied in real time during surgery to increase contrast and improve the visualization of structures. Several studies have reported improvements in visualization with digital images for both cataract surgery and epiretinal membrane staining during surgery [14,15]. Here, we describe the feasibility of using digital visualization and its contribution to the enhancement of MP visualization and analysis in a prospective series of macular surgery interventions. 

## 2. Materials and Methods

We prospectively studied patients undergoing macular surgery at Guillaume de Varye Hospital (Bourges, France) between September and October 2022. The types of surgery performed included vitreomacular traction, macular hole surgery, and epiretinal membrane surgery. This study was approved by the ethics committee of our institution, and informed consent was obtained from the patients before inclusion. The study was performed in accordance with the Declaration of Helsinki. 

### Technique

All interventions were performed by the same experienced surgeon (O.S.) using a 3D digital visualization system (NGENUITY^®^, Alcon, Fort Worth, TX, USA) consisting of a high-dynamic range video camera connected to a microscope (Lumera 700 Carl Zeiss Meditec, Jena, Germany) in place of the oculars. The color temperature of the halogen lamp used was 3400 Kelvin. The iris diaphragm of the digital video camera was set to approximately 30% open for 3D surgery.

All operations were performed under general anesthesia, with Ultra High 10,000 cpm cut-rate Alcon Constellation^®^ 25-gauge vitrectomy probes. After vitrectomy, epiretinal membrane and internal limiting membrane peeling was performed under contact lens viewing (30° FCI^®^), with DOUBLEDYNE^®^ (Horus pharma, Saint-Laurent-du-Var, France) blue dye. The intensity of the light probe was set at 20% of the maximum value.

Two optimized 3D digital filters were customized before the start of this study, based on the surgeon’s assessment of the enhancement of MP visualization. The first of these filters greatly improved MP visualization and involved the use of the following settings for the color filters: cyan-red filter (60% instead of 100%), magenta-green filter (40% instead of 100%), and blue-yellow filter (70% instead of 100%). The other image parameters remained unchanged, with settings of 47.80 for brightness, 54.90 for contrast, 1.20 for gamma, and 2 for hue.

In the second optimized filter, the enhancement of MP visualization was less marked, but contrast was improved, facilitating safe ILM peeling. This second filter differed from the first in terms of green-magenta saturation (110% instead of 40%). OCT scans were performed on the day of surgery in all patients without gas or air tamponade. 

The variables evaluated were MP enhancement, MP regularity, and enhancement of the pigmented epiretinal tissue associated with the lamellar or macular hole. 

## 3. Results

We included 21 eyes from 21 patients undergoing vitrectomy for macular surgery. Five of these patients underwent vitreomacular traction, five underwent surgery for macular holes, nine underwent macular epiretinal membrane surgery, and two underwent surgery for lamellar holes. The mean age of the patients was 70.0 ± 4.7 years, and 12 of the 21 eyes were from male patients. All macular holes were successfully closed, as demonstrated by OCT performed three weeks after surgery following gas resorption. No ocular complications occurred in any of the interventions.

### Tissue Color Modifications

Filter 1 provided the strongest enhancement of MP visualization, with this pigment appearing green and slightly fluorescent. The retina took on a magenta tint, and its surface was less clearly defined than with standard color (Figure 1).

Filter 2 provided weaker enhancement of the MP than the first filter. It induced the retina to take on a cyan tint. The retina surface was visualized at least as sharply as with standard color (Figure 1).

In vitreomacular surgery, the filters facilitated the assessment of MP integrity at the end of surgery. Disruption of the MP was noted in one case. This patient already displayed imminent macular hole formation and was treated with SF6 gas tamponade. In the other cases, the regularity of the MP was conserved at the end of surgery with the use of filter 1. OCT on the day after surgery confirmed the absence of a macular hole. In one patient, the inner retinal roof covering the pseudocyst was lost in the previous vitreomacular traction zone; the outcome in this patient was good (Figure 2).

In full-thickness macular hole surgery, the filters highlight the disruption of the MP inside the hole area. One patient had epiretinal proliferation associated with the macular hole; this proliferation was non-uniform, with one part appearing green, while the other part did not (Figure 3). In the other macular holes without epiretinal proliferation, the peeled tissue around the hole was homogeneous without enhancement with digital MP filters.

For lamellar holes, the filters revealed continuity of the MP in the lamellar hole area. They enhanced the visualization of the epiretinal proliferation associated with these lamellar holes. In epiretinal membrane surgery, the filters showed that the integrity of the MP was maintained at the end of surgery in all cases (Figure 1). The peeled ILM over the fovea was homogeneous without enhancement with digital MP filters.

## 4. Discussion

High concentrations of MP are normally present in the macula area, and various aspects of MP disruption in retinal diseases have been described. In clinical practice, MP can be enhanced and evaluated using the two-wavelength fundus autofluorescence technique [5,6]. Here, we provide the first report of the digital enhancement of MP visualization during surgery.

The main features of the optimized settings used in our study to enhance MP visualization were increases in cyan and yellow saturation and the use of a magenta-green filter.

Increasing cyan color saturation in the cyan-red channel decreased the color temperature of the warm colors, close to red, in the image [16], thereby highlighting the xanthophyll pigment. Yellow color saturation was also increased to highlight the yellowish color of the MP. Blending of cyan and yellow colors typically results in a green color, which we observed intraoperatively with the use of these cyan and yellow filters. 

The magenta-green filter was the third component of the optimized settings, and its use differentiated between filters 1 and 2 in this study. Increasing magenta color saturation (filter 1) resulted in the MP appearing clearly green and slightly fluorescent. In terms of color temperature, magenta is complementary to green, with complementary colors defined as the colors with the largest contrast between them [17]. Decreasing magenta saturation (filter 2) decreased MP enhancement but increased the quality of the retinal surface visualization. This filter seems to provide a good compromise between MP enhancement and good retinal surface visualization.

Consequently, the two filters could be used during different steps in retinal surgery. Filter 1 could be used to assess MP at the end of surgery, whereas filter 2 is suitable for use even during retinal peeling. In clinical surgical practice, these filters could be applied in macular surgery, mostly for the epiretinal tissue associated with lamellar or macular holes and for vitreomacular traction operations.

The pigmented epiretinal tissue associated with lamellar or macular holes is composed of Müller cells. Preservation of this epiretinal proliferative tissue is recommended to ensure a successful surgical outcome. Indeed, Shiraga et al. [9] suggested that better clinical outcomes are obtained if the “thick ERM with MP” is preserved than if it is removed. 

However, the MP may be difficult to see clearly during surgery with standard visualization because it has a yellow color not dissimilar to the orange color of the retina, so the degree of contrast between the MP and the retina is, therefore, low. Filter 2 proved very useful for highlighting the associated pigmented epiretinal tissue in cases of lamellar and macular holes, facilitating the preservation of this pigmented tissue and its positioning within the hole at the end of surgery. 

In vitreomacular traction surgery, there is a risk, albeit rare, of a full-thickness macular hole occurring when the posterior vitreous is detached from the fovea. The presence of an iatrogenic macular hole modifies the surgical approach, as the surgeon should perform additional internal limiting membrane peeling and gas tamponade. Filter 1 was useful for assessing MP integrity at the end of surgery, and it highlighted the presence of a full-thickness macular hole in one case. Several abnormalities of MP distribution have been described at various stages of macular hole development [6]. However, this filter should be evaluated in a large series of vitreomacular traction operations using simultaneous correlation with intraoperative OCT. Currently, intraoperative OCT, which provides real-time retinal visualization, is the gold standard tool for the detection of this type of surgical complication, but it is not readily available at many centers. MP analysis with digital filters represents another biomarker that can be measured intraoperatively, potentially serving as a predictive factor for postoperative outcome in macular diseases. Jaggi et al. recently demonstrated that postoperative MP fluorescence was predictive of long-term postoperative outcome in cases of retinal detachment [18].

The digital filter settings described here can be modified slightly according to the surgeon’s personal preferences in each individual case of retinal surgery, as a function of retinal pigmentation. Image quality can also be modified with the NGENUITY digital color system by changing other parameters, such as contrast, hue, brightness, gamma, and color saturation. We recently developed a new optimized filter that provides a very high degree of MP enhancement (Appendix A). We used the settings of filter 1 to customize this filter, with an increase in color temperature and a decrease in the brightness of the image (Figure 4). Increasing the color temperature adds a warm tone to the image, making the colors on the image appear warmer. Decreasing the brightness of the image also enhanced the contrast between the MP zone and the macular zone without MP. This effect is probably due to the fluorescence of MP because, despite the decrease in image brightness, the MP remains luminous and fluorescent [19]. 

Currently, no digital visualization system can quantify MP in real time. In the near future, it will become possible to modify intraoperative images in the visualization software postoperatively to enhance visualization so that new filters can be added efficiently. Ultimately, artificial intelligence will permit intraoperative analysis of many biomarkers that include MP, with the hope of improving surgical decision making. 

This study is limited by the relatively small number of cases in our series. Our findings should be evaluated in a larger series and in other diseases using simultaneous correlation with intraoperative OCT.

## 5. Conclusions

In conclusion, we report optimized digital filters for use in a digital 3D visualization system for the enhancement of MP visualization during surgery. These filters enhanced the MP and pigmented epiretinal tissue visualization to produce a measurable and reproducible intraoperative biomarker; however, it should be evaluated in a larger series of eyes with macular diseases requiring surgical treatment.

## Figures and Tables

**Figure 1 jcm-12-02300-f001:**
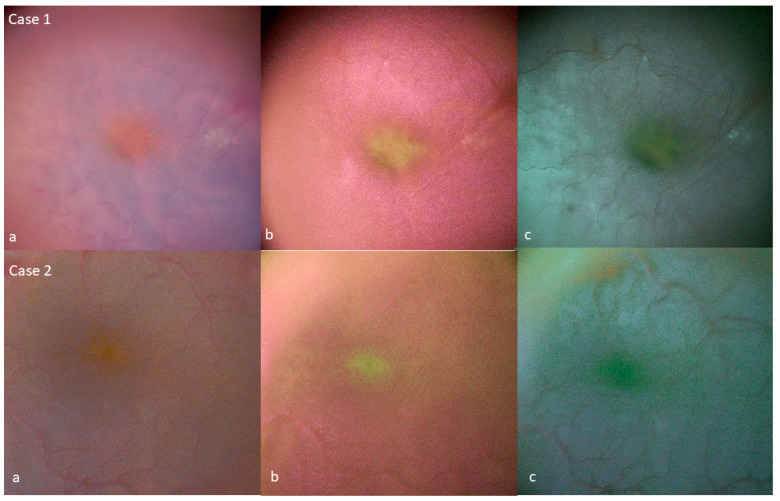
Enhancement of the visualization of MP at the end of surgery for epiretinal membrane surgery. In two different cases (**a**) Standard color, (**b**) filter 1, (**c**) filter 2.

**Figure 2 jcm-12-02300-f002:**
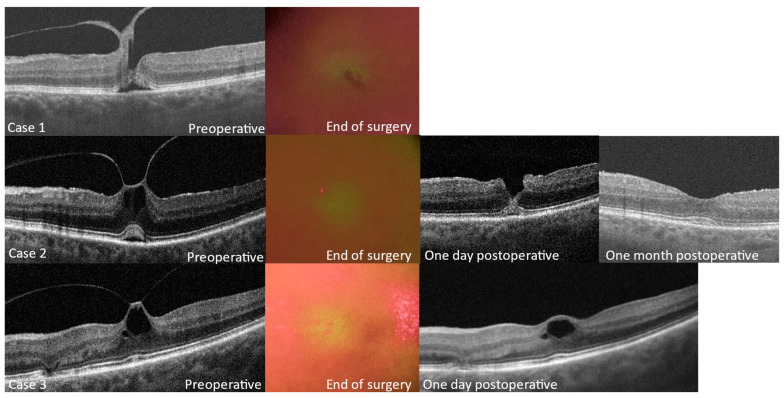
Analysis of macular pigmentation with filter 1 at the end of surgery in cases of vitreomacular traction. Preoperative and postoperative scans of patients are presented except for case 1 who had gas tamponade. In case 1, the MP appears to have a defect. In cases 2 and 3, the MP has a regular appearance. Note small artifacts in case 3 related to triamcinolone deposits.

**Figure 3 jcm-12-02300-f003:**
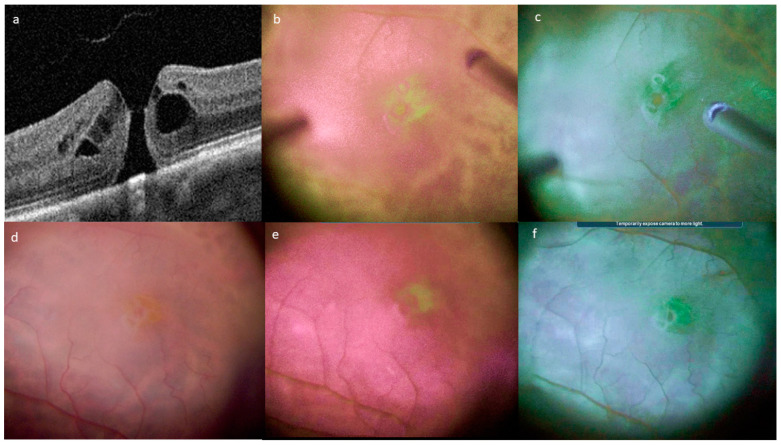
Enhanced visualization of the pigmentation of epiretinal tissue associated with a macular hole. (**a**) Preoperative OCT. Enhancement of pigmented epiretinal tissue with filter 1 (**b**) and filter 2 (**c**) during peeling. Standard color view (**d**) after positioning of the pigmented epiretinal tissue within the hole after. View after filter 1 (**e**) and filter 2 enhancement (**f**).

**Figure 4 jcm-12-02300-f004:**
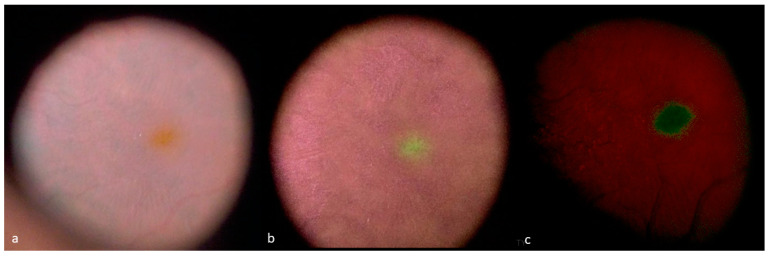
Enhancement of MP visualization in a case of epiretinal membrane surgery. (**a**) Standard colors, (**b**) filter 1, (**c**) high-enhancement of MP filter; additional changes were applied to filter 1: 32.02 for brightness, 69,70 for contrast, 1.40 for gamma, 60 for hue, 50% for color saturation, and 6000 kelvin for color temperature.

## Data Availability

The data that support the findings of this study are available from the corresponding author, S.O. upon request.

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
