# Peer review of "Operative Digital Enhancement of Macular Pigment during Macular Surgery"

_jcm, 2023, doi:10.3390/jcm12062300_

Round 1

Reviewer 1 Report

Dear Authors,

Congratulations for your work! The 3D digital visualization system has become quite present in VR surgeons' OR and expanding the digital possibilities of these magnificent feature is highly desired by everyone involved in.

I am suggesting just a few minor changes regarding the editing of the text:

1. Page 2 line 76 you should mention for the commercial name of doubledyne the provider also.

2. Page 5 line 129 you should correct the legend of the Figure 3, because the "f" image is missing

3. Page 6 line 188 - you are mentioning on figure 4 c - high-enhancement with ST MP filter, but you are not defining anywhere what "ST" means. Please do it accordingly.

Best regards!

Author Response

Reviewer 1:

Congratulations for your work! The 3D digital visualization system has become quite present in VR surgeons' OR and expanding the digital possibilities of these magnificent feature is highly desired by everyone involved in.

Thank you for this very positive comment, your interest in our study, and your highly pertinent suggestions. Digital visualization can, indeed, significantly improve the visualization of structures during surgery, and different filters can be optimized for particular surgical situations (retina surgery, cataract, glaucoma, etc. ).

I am suggesting just a few minor changes regarding the editing of the text:

  1. Page 2 line 76 you should mention for the commercial name of doubledyne the provider also.

Thank you for this suggestion. We now indicate the supplier of the doubledyne dye in the revised version of the manuscript.

< DOUBLEDYNE ® (Horus pharma) blue dye>

  1. Page 5 line 129 you should correct the legend of the Figure 3, because the "f" image is missing

Thank you for this comment. The image for “f” is indeed missing and the legend has been corrected accordingly in the revised version of the manuscript.

  1. Page 6 line 188 - you are mentioning on figure 4 c - high-enhancement with ST MP filter, but you are not defining anywhere what "ST" means. Please do it accordingly.

The “ST” refers to the initials of the two authors (Sandali and Tahiri) who developed the filter settings. This abbreviation has been removed from the revised version of the manuscript.

Thank you again for very helpful comments.

Reviewer 2 Report

The aim of the manuscript submitted by the authors is to describe a method to digitally enhance the intra-operative visualization of macular pigment during macular surgery. 

The work presents several issues that must be addressed: 

-        Introduction: it should be organized in a clearer fashion (especially lines 42-51)

-        Results: “Filter 2 gave a weaker enhancement of the MP than the first filter. It induced the retina to take on a cyan tint. The retina surface was visualized as sharply as with standard color”  Could you please highlight the pros of using filter 2 instead of no-filter? What is the rationale of changing colors parameters if there is no improvement in terms of visualization? 

-        Looking at figure 1, it looks like the use of filter 2 mitigates the visibility of doubledyne blue dye.  

-        Page 3, lines 112-113: “However, one patient developed a superficial tear of the inner retinal surface in the previous vitreomacular traction zone, for which the outcome was good (Figure2).” The aforementioned superficial tear is not visible in figure 2.

-        The results could be presented more extensively

-        Discussion looks very confusing, providing incomplete information and should be extensively reviewed. It should focus more on the application of the described findings in the clinical practice and how it could affect prognosis. Authors should better describe pros and cons of the technique and further discuss the limitations of the technique.

Author Response

Responses to the comments of reviewer 2:

Thank you for your interest in our study and for your pertinent suggestions, which have helped us to improve the quality of our paper. 

Changes to the manuscript have been noted in this document after each comment and also in the main manuscript using MS Word/LaTeX Track Changes.

1/ Introduction: it should be organized in a clearer fashion (especially lines 42-51)

 We agree with the reviewer. The introduction section has been modified so as to increase its clarity for readers.

In these two paragraphs (lines 42-51), we report potential applications of MP analysis during retinal surgery. After the definition of “macular pigment”, we have added a sentence to introduce these two paragraphs.

< Aspects of MP disruption have been described in surgical retinal diseases, mostly in vitreomacular traction and epiretinal proliferation associated with lamellar or macular hole diseases. Indeed, Obana et al. recently described the appearance of the MP in eyes with different stages of macular hole [6]. They observed a lack of MP within the hole, in an area corresponding to an outer plexiform layer defect. MP analysis may, therefore, be of interest in vitreomacular traction surgery, for the detection of iatrogenic macular holes.

MP has been reported to be present in the epiretinal proliferation associated with lamellar or macular holes [7]. Gentle handling, with preservation of the epiretinal proliferative tissue, is recommended to ensure a successful outcome of surgery for these diseases [8,9]. However, in surgical practice with standard visualization, it may be difficult to obtain a clear view of the MP, the yellowish color of which is close to the orange color of retina, resulting in low contrast between them.>

2/ Results: “Filter 2 gave a weaker enhancement of the MP than the first filter. It induced the retina to take on a cyan tint. The retina surface was visualized as sharply as with standard color”  Could you please highlight the pros of using filter 2 instead of no-filter? What is the rationale of changing colors parameters if there is no improvement in terms of visualization? 

 Thank you for this pertinent comment. Filter 2 gave a weaker enhancement than filter 1, but a higher resolution than filter 1 with a persistent MP enhancement in comparison with no filter. Filter 2 was therefore useful even during retinal peeling. Filter 2 was particularly useful for highlighting associated pigmented epiretinal tissue in cases of lamellar and macular holes, facilitating the preservation of this pigmented tissue and its positioning within the hole at the end of surgery.

Filter 1 gave the greatest enhancement of MP, allowing detailed analysis of the MP at the end of surgery, but it was not suitable for use during retinal peeling.  This filter could therefore be used for MP analysis at the end of surgery, particularly for macular traction disease, or for the detection of associated epiretinal tissue during surgery for macular holes.

Discussion section of the revised version of the manuscript:

“The two filters could be used during different steps in retinal surgery. Filter 1 could be used to assess MP at the end of surgery, whereas filter 2 is suitable for use even during retinal peeling. Filter 2 proved very useful for highlighting associated pigmented epiretinal tissue in cases of lamellar and macular holes, facilitating the preservation of this pigmented tissue and its positioning within the hole at the end of surgery.”

3/ Looking at figure 1, it looks like the use of filter 2 mitigates the visibility of doubledyne blue dye.  

 Thank you for this comment. Indeed, in figure 1 (case 1), filter 2 may appear to mitigate the visibility of the doubledyne blue dye. As stated in the legend to figure 1, the images were recorded after ILM peeling. There was, therefore no true staining of the retinal surface in this case. This misleading appearance of the blue coloration of the retina can be explained by the residual blue dye in the vitreous cavity after restaining at the end of the surgery.  Filter 2 consists principally of a cyan filter increasing the visibility of tissues of a color close to cyan. In routine surgical practice, I use a cyan filter to enhance the blue color of the ILM.

4/ Page 3, lines 112-113: “

However, one patient developed a superficial tear of the inner retinal surface in the previous vitreomacular traction zone, for which the outcome was good (Figure2).” The aforementioned superficial tear is not visible in figure 2.

Thank you for this important comment. Indeed, it highlights the possible misunderstanding of the formulation “superficial tear of the inner retinal surface” by readers.

Vitreomacular traction-related pseudocysts take several weeks to disappear after surgery without tamponade, as in case 3, shown in figure 2. In case 2, the OCT on the day after surgery clearly showed that the pseudocyst had retained its shape but that the roof of the pseudocyst had been lost.

The “superficial tear of the inner retinal surface” formulation has been modified in the results section, to increase clarity.

“However, in one patient, the inner retinal roof covering the pseudocyst was lost in the previous vitreomacular traction zone; the outcome in this patient was good (Figure 2).”

5/ The results could be presented more extensively

Thank you for this comment.

Indeed, the basic information for the 21 eyes included in the study was not initially provided. These data are now included in the results section of the study.

We included 21 eyes from 21 patients undergoing vitrectomy for macular surgery. Five of these patients underwent vitreomacular traction, five underwent surgery for macular holes, nine underwent macular epiretinal membrane surgery, and two underwent surgery for lamellar holes. The mean age of the patients was 70.0 ± 4.7 years, and 12 of the 21 eyes were from male patients. All macular holes were successfully closed, as demonstrated by OCT performed three weeks after surgery, following gas resorption. No ocular complications occurred in any of the interventions.”

After this section, modifications to the color of the retina with filters 1 and 2 are described. MP analysis is then described for each of the diseases included in the study.

Additional data were provided in the result section.

In vitreomacular surgery, the filters facilitated the assessment of MP integrity at the end of surgery. A disruption of the MP was noted in one case. This patient already displayed imminent macular hole formation and was treated by SF6 gas tamponade. In the other cases, the regularity of the MP was conserved at the end of surgery, with the use of filter 1. OCT on the day after surgery confirmed the absence of a macular hole. However, in one patient, the inner retinal roof covering the pseudocyst was lost in the previous vitreomacular traction zone; the outcome in this patient was good (Figure 2).

In full-thickness macular hole surgery, the filters highlight the disruption of the MP within the hole area. One patient had an epiretinal proliferation associated with the macular hole; this proliferation was non-uniform, one part appearing green whereas the other part did not (Figure 3). In the other macular holes without epiretinal proliferation, the peeled tissue around the hole was homogeneous without enhancement with digital MP filters.

For lamellar holes, the filters revealed a continuity of the MP in the lamellar hole area. They enhanced visualization of the epiretinal proliferation associated with these lamellar holes.

In epiretinal membrane surgery, the filters showed that the integrity of the MP was maintained at the end of surgery, in all cases (Figure 1). The peeled ILM over the fovea was homogeneous without enhancement with digital MP filters.”

6/ Discussion looks very confusing, providing incomplete information and should be extensively reviewed. It should focus more on the application of the described findings in the clinical practice and how it could affect prognosis. Authors should better describe pros and cons of the technique and further discuss the limitations of the technique.

Thank you for this comment. The discussion section has been modified and restructured in the revised version, to increase its clarity for readers.

The first part of the discussion is a technical description of the color modifications achieved with these optimized filters and the enhancement of MP. Thereafter, as suggested by reviewer 2, we describe in more detail the applications of these MP filters in clinical practice, the pros and cons of this technique, and its limitations.

The main paragraphs modified in the new version of the discussion of our manuscript are provided below:

< This filter provided a good compromise between MP enhancement and good retinal surface visualization. Consequently, the two filters could be used during different steps in retinal surgery. Filter 1 could be used to assess MP at the end of surgery, whereas filter 2 is suitable for use even during retinal peeling.

In clinical surgical practice, these filters could be applied in macular surgery, mostly for the epiretinal tissue associated with lamellar or macular holes and for vitreomacular traction operations.

The pigmented epiretinal tissue associated with lamellar or macular holes is composed of Muller cells. The preservation of this epiretinal proliferative tissue is recommended to ensure a successful surgical outcome. Indeed, Shiraga et al. [9] suggested that better clinical outcomes are obtained if the “thick ERM with MP,” is preserved than if it is removed.

However, the MP may be difficult to see clearly during surgery with standard visualization because it has a yellow color not dissimilar to the orange color of retina, and the degree of contrast between the MP and the retina is, therefore, low.

Filter 2 proved very useful for highlighting associated pigmented epiretinal tissue in cases of lamellar and macular holes, facilitating the preservation of this pigmented tissue and its positioning within the hole at the end of surgery.

In vitreomacular traction surgery, there is a risk, albeit rare, of a full thickness macular hole occurrence when the posterior vitreous is detached from the fovea. The presence of an iatrogenic macular hole modifies the surgical approach as the surgeon should perform additional internal limiting membrane peeling and gas tamponade. Filter 1 was useful for assessing MP integrity at the end of surgery, and it highlighted the presence of a full-thickness macular hole in one case. Several abnormalities of MP distribution have been described at various stages of macular hole development [6]. However, this filter should be evaluated in a large series of vitreomacular traction operations, with simultaneous correlation with OCT findings during surgery. Indeed, intraoperative OCT providing real-time retinal visualization is the gold standard tool for the diagnosis of this complication of surgery.

MP analysis with digital filters may pave the way for future studies increasing our knowledge, even if they do not necessarily improve surgical outcomes. Indeed, MP disturbance could be evaluated as a predictive factor for postoperative outcome in macular diseases.  Jaggi et al. recently demonstrated that postoperative MP fluorescence was predictive of long-term postoperative outcome in cases of retinal detachment [18].

One possible limitation of this study is that we did not evaluate MP enhancement with objective contrast measurement parameters. However, that was not the aim of this study, which was designed as a descriptive report study evaluating the feasibility of digitally enhancing the MP during macular surgery. Another limitation of the study is the limited cases in our series. Our findings should be evaluated in larger series and in other diseases, with simultaneous correlation with intra operative OCT.

Changes to the manuscript have been noted in the main manuscript using MS Word/LaTeX Track Changes.

We again thank reviewer 2 for these pertinent comments, which have helped us to improve the quality of the manuscript.